# Natural Products for Melanoma Therapy: From Traditional Medicine to Modern Drug Discovery

**DOI:** 10.3390/plants14060951

**Published:** 2025-03-18

**Authors:** Soojin An, Jeongeun An, Dain Lee, Han Na Kang, Sojin Kang, Chi-Hoon Ahn, Rony Abdi Syahputra, Rosy Iara Maciel A. Ribeiro, Bonglee Kim

**Affiliations:** 1College of Korean Medicine, Kyung Hee University, Seoul 02447, Republic of Korea; soojin0614@khu.ac.kr (S.A.); lisaan0508@gmail.com (J.A.); leedain1342@khu.ac.kr (D.L.); bmb1994@khu.ac.kr (S.K.); ach2565@naver.com (C.-H.A.); 2Department of Pathology, College of Korean Medicine, Kyung Hee University, Hoegidong Dongdaemun-gu, Seoul 02447, Republic of Korea; 3KM Convergence Research Division, Korea Institute of Oriental Medicine, Daejeon 34054, Republic of Korea; khn3110@kiom.re.kr; 4Department of Biological Sciences, State Islamic University of Sunan Kalijaga (UIN Sunan Kalijaga), Yogyakarta 55281, Indonesia; rony@usu.ac.id; 5Laboratory of Experimental Pathology, Federal University of São João del Rei-CCO/UFSJ, Divinópolis 35501-296, Brazil; rosy@ufsj.edu.br

**Keywords:** melanoma, natural products, antioxidant, anti-cancer, apoptosis, reactive oxygen species (ROS), angiogenesis, metastasis

## Abstract

Melanoma, a type of skin cancer originating from melanocytes, represents a significant public health concern according to the World Health Organization. It is one of the most commonly diagnosed cancers worldwide, particularly affecting populations in Europe and North America, with an increasing incidence in Asia. The rise emphasizes the need for diversified treatment approaches. Conventional treatments for melanoma, including immunotherapy, chemotherapy, and targeted therapies like the FDA-approved Opdivo and Relatlimab, often come with severe side effects and high relapse rates. Consequently, natural products have gained considerable attention for their potential to enhance therapeutic outcomes and reduce adverse effects. This systematic review evaluates the anti-cancer properties of natural products against melanoma, examining 52 studies from PubMed and Google Scholar. Our analysis focuses on the antioxidant, anti-angiogenesis, anti-metastatic, and apoptosis-inducing activities of these compounds, also discussing the regulatory factors involved. The findings advocate for intensified research into natural products as complementary agents in melanoma treatment, aiming to improve efficacy and patient quality of life. Further in vitro, in vivo, and clinical trials are essential to validate their effectiveness and integrate them into standard care protocols.

## 1. Introduction

The skin, the largest organ of the human body, is susceptible to various stressors such as sunlight, injury, and infection [1]. As of 2024, melanoma ranked 17th in incidence and 22nd in mortality rates among cancers [2]. According to the Global Cancer Observatory, melanoma caused 9.7 million deaths among 20 million new cases of cancer in people around the globe in 2022 [3]. In addition, it is expected to affect 21.3 million in 2025, along with demographic changes [4]. This points to one of the crucial takeaways and motivations of this research, which is that melanoma has been slowly showing its prevalence in Asia in recent years [5].

Asian patients with melanoma have a 27% higher mortality rate compared to other Caucasian patients [5]. The National Institutes of Health (NIH) National Cancer Institute claims that superficially spreading melanoma makes up 70% of melanoma cases, and it is the most common type of melanoma that many are familiar with. According to the National Institutes of Health (NIH) National Cancer Institute, melanoma is a disease that arises from the malignant transformation of melanocytes. Melanocytes are responsible for producing melanin, the pigment that gives skin its color. It commonly develops on sun-exposed skin, and it can also occur in less visible areas, such as the eyes, nose, and throat. Although the exact cause of melanoma is unclear, a major contributing factor is believed to be exposure to ultraviolet (UV) radiation from sunlight and artificial sources like tanning beds. People subjected to prolonged UV exposure have a higher likelihood of developing melanoma. Excessive sun exposure can also contribute to skin pigmentation changes. Some plant-derived extracts have been studied for their potential effects on melanin production and UV-induced skin damage [6]. Melanoma is most common in individuals in their mid-40s and above, and is known for its high metastatic potential, often growing silently and undetected, making early intervention crucial. Fortunately, when detected early, melanoma can be effectively treated [7].

The World Health Organization (WHO) identifies melanoma as one of the most common endocrine-rooted cancers [8]. Until recently, melanoma was predominantly diagnosed in populations in the United States and Europe. However, according to the Global Cancer Observatory, in 2020, over 10,000 new cases were reported in Asia, following trends seen in Western countries. Extrapolating from these data, it is possible that melanoma will become one of the more frequently diagnosed cancers in Asian countries. As global temperatures continue to rise due to climate change, people will be increasingly exposed to higher levels of UV radiation, which could lead to an increase in melanoma incidence.

Melanoma has historically been more common in men than in women [9]. By 2020, more than 52,000 melanoma-related deaths occurred globally, regardless of gender. Additionally, medical data published by the Mayo Clinic in April 2023 emphasized that men, particularly those over 50, are at a higher risk of melanoma, with their risk increasing as they age. As melanoma continues to emerge as a serious cancer, preventive measures and treatment strategies must be thoroughly researched.

There are several methods to treat melanoma, including immunotherapy, targeted therapies, chemotherapy, radiation therapy, and surgery [10]. Novel FDA-approved drugs, such as Opdivo (nivolumab) and Relatlimab, are available, and early detection could drastically increase the 5% 5-year-overall survival rate to 50% [11,12]. However, melanoma is notorious for its tendency to relapse, even after treatment. As noted by Berthenet and Weber, melanoma cells often exhibit limited apoptosis during chemotherapy [13]. However, when pitavastatin, an FDA-approved drug, has been used alongside chemotherapy, active apoptotic activity has been observed compared to chemotherapy alone [14].

Although there are other monoclonal antibodies, besides those mentioned above, targeting melanoma, this review emphasizes the role and potential of traditional medicine in the field of cancer. Adjuvant treatment with the natural products often used in Traditional Chinese Medicine (TCM) and Traditional Korean Medicine (TKM) may reduce side effects, overcome drug resistance and increase survival rates [15], resulting in improved treatment outcomes in melanoma patients. Chemotherapy and immunotherapy cause deleterious effects on the human body. Various medicinal plants have been traditionally used in treating skin diseases and inflammation, and studies have reported that certain plant-derived compounds exhibit anticancer and anti-inflammatory properties [16]. Such therapies not only target cancer cells, but also affect healthy cells, contributing to side effects like weight loss and fatigue. If the synergistic effects of natural products with skin cancer therapies could reduce the number of chemotherapy sessions required, they could improve patients’ quality of life, the key goal in modern medicine [17]. Single TCM and TKM elements and pure natural compounds, along with their efficacy against melanoma cells, should be investigated to prompt future studies on adjuvant therapies. The authors strongly believe that natural products have great potential use in treating melanoma, a cancer prone to relapse, and should be further investigated in the medical field.

## 2. Methods

### 2.1. Search Strategy

Research papers addressing natural products that are effective in targeting melanoma were collected from February 2023 to March 2024 using databases PubMed, Google Scholar, Web of Science, and Scopus, using different combinations of keywords, such as the following: skin cancer, melanoma, natural products, herbal medicine, botanical medicine, plant extracts, antioxidant, anti-cancer, apoptosis, reactive oxygen species (ROS), angiogenesis and metastasis.

### 2.2. Study Selection Criteria

The selection of studies was based on the following inclusion criteria: (1) studies regarding the efficacy of natural products in any dosage form on melanoma cell or patients; (2) preclinical (in vivo, in vitro) studies; (3) isolated natural product; (4) original research articles; (5) articles written in English; (6) articles published within the last 10 years. The exclusion criteria were as follows: (1) articles regarding synthetic compounds; (2) systematic reviews, meta-analyses, review articles, conference papers and case reports; (3) articles in which the full text is unavailable. The resulting articles were revised to manually identify and exclude the articles that did not meet the criteria above. To identify articles of interest, all the articles retrieved from two databases were manually reviewed and checked for duplication.

### 2.3. Data Extraction

The obtained information included the author’s name, the publication year, the disease targeted, the extracted natural product, the cell line/animal model type, the dose and duration of extract administered, and the mechanisms of action (if reported) (Figure 1).

## 3. Anti-Melanoma Effect of Natural Products

### 3.1. Apoptosis and Natural Products

Natural products ranging from extracts to single compounds, such as flavonoids, exhibit apoptotic effects on malignant melanoma cells (Table 1 and Figure 2).

Khan et al. reported the apoptotic effect of porcupine bezoar on A375 melanoma cells [18]. Porcupine bezoar at a dose of 3 µg/mL for 72 h had the highest apoptotic effect on A375 cells. The natural product exhibited apoptosis by the mitochondria apoptosis pathway, down-regulating Bcl-2, and up-regulating Bax, cyt-c, c-caspase-3, and c-capase-9. Srivastava et al. investigated the apoptotic effects of curcumin and quercetin on the A375 malignant melanoma cell line [19]. Curcumin and quercetin, at a dose of 1.5, 3.1, 6.3, 12.5, 25, or 50 μM for 24 h, resulted in the activation of apoptotic enzymes in a concentration-dependent manner. Apoptosis was induced through the down-regulation of DVL2, β-catenin, cyclin D1, Cox2, Axin2, Bcl-2, c-caspase-3, and c-caspase-7. Netala et al. assessed the apoptotic effects of the fungus *Pestalotiopsis microspore* on the B16F10 melanoma cell line [20]. *Pestalotiopsis microspore* at doses of 10, 20, 30, 40, 50, 100, 150, and 200 µg/mL induced apoptosis through the up-regulation of reactive oxygen species (ROS). And Ohkubo et al. demonstrated the apoptotic effects of aged garlic extract (AGE) on the M14 WT melanoma cell line [21]. Doses of AGE ranging from 0 to 10 mg/mL, exposed to cells for 24, 48, and 72 h, led to the up-regulation of proteins Cytc, Smac, and Diablo in a dose-dependent manner. Barros et al. highlighted the apoptotic effects of cashew gum on B16F10 cells and in C57BL/6 mice [22]. Doses of 3.125 to 100 μg/mL for 72 h activated apoptotic proteins in a dose-dependent manner. In vivo, C57BL/6 mice were intraperitoneally injected with 50 and 100 mg/kg of cashew gum for 14 consecutive days, which down-regulated γH2AX, ultimately inducing apoptosis in melanoma cells. Al Qathama et al. reported the apoptotic effects of Saudi medicinal plants on A375 and B16F10 melanoma cell lines [23]. *C. decidua, and C. edulis,* and *H. tuberculatum,* at a dose of 0.45 to 15 µg/mL for 48 h, induced apoptosis through the down-regulation of cleaved caspase-3 and cleaved caspase-7. Danciu et al. illustrated the apoptotic effects of chamomile, parsley and celery on the A375 melanoma cell line [24]. Dosages of chamomile, parsley and celery ranging from 10 to 60 μg/mL exposed to the melanoma cell line for 72 h resulted in the activation of various proteins in a dose-dependent manner. Apoptosis was induced by up-regulating c-casapse-3 and down-regulating IL-10. Ferreira et al. reported the apoptotic effect of *Conyza bonariensis* (L.) on SK-MEL-28, a melanoma cell line [25]. Exposing SK-MEL-28 to 20 μg/mL of *Conyza bonariensis* (L.) plant extract for 48 h induced apoptosis by activating ROS. Huang et al. investigated the apoptotic effects of *Curcuma longa* Linn extract on A2048 melanoma cells [26]. Doses of 5 to 25 μΜ of the extract, consisting of curcumin, demthoxycurcumin and bismethoxycurcumin, for 24 and 48 h resulted in apoptosis, coherent with the results of Srivastava et al. Also, Figueiredo et al. studied the apoptotic effects of curzerene extracted from *Eugenia uniflora*. SKMEL-19 cells were treated with 5 and 10 μM for 72 h [27]. This limited cell migration and enhanced chromatin condensation, promoting apoptosis. Kaushik et al. demonstrated the apoptotic effects of *Hibiscus* and *Cinnamomum* on the G361 melanoma cell line [28]. *Hibiscus* and *Cinnamomum,* at a dose of 62.5 to 1000 μg/mL for 48 h, caused apoptosis in a dose-dependent manner. Apoptosis was stimulated through the major metabolites found in *Hibiscus* and *Cinnamomum*. Merecz-Sadowska et al. demonstrated the apoptotic effect of *Leonotis nepetifolia* root extract on the A375 melanoma cell line [29]. *Leonotis nepetifolia,* at a dose of 1500 mg/mL for 24 h, had the highest apoptotic effect on A375 cells. Applying *Leonotis nepetifolia* root extract for 24 h resulted in the up-regulation of TP53 and p-53 in a concentration-dependent manner. Thaichinda et al. highlighted the apoptotic effects of pine bark on the human malignant A375 melanoma cell line [30]. *Pinus maritima,* at a dose of 5 to 100 µg/mL for 48 h, caused apoptotic activity by stimulating various proteins in a dose-dependent manner. Apoptosis was induced through the up-regulation of ROS and c-caspase 3 and the down-regulation of MMP-9. Bożek et al. demonstrated the apoptotic effects of *Piptoporus betulinus* ethanolic extracts on A375 cells [31]. Doses of 2.5 to 10 μL/mL for 24 h resulted in the up-regulation of caspase-3 activity. Oliveira et al. illustrated the apoptotic effects of Portuguese propolis, *Apis mellifera* L., on BRAF, A375, and WM9 melanoma cell lines [32]. Dosages of *Apis mellifera* L. ranging from 5 to 60 μg/mL for 72 h resulted in the activation of various proteins in a dose-dependent manner. Apoptosis was induced through the up-regulation of ROS, Bax, c-caspase-3, c-caspase-9 and p53, and the down-regulation of Bcl-XL and Bcl-2. Prasedya et al. demonstrated the apoptotic effects of red macroalgae, *Gelidium latifolium*, on the B16F10 malignant melanoma cell line [33]. Red macroalgae at a dose of 10 to 200 μg/mL for 72 h resulted in the activation of apoptotic enzymes in a concentration-dependent manner. Apoptosis was induced through the up-regulation of p53, Bax and Bak and the down-regulation of Bcl-2. Chatti et al. reported the apoptotic effect of *Rhamnus alaternus* on the B16F10 melanoma cell line [34]. *Rhamnus alaternus* at a dose of 20 μg/mL for 48 h had the highest apoptotic effect on B16F10 cells. *Rhamnus alaternus* aided in inhibiting cell proliferation and cell cycle arrest by damaging DNA. In addition to the proliferating effects of *Rhamnus alaternus*, apoptosis was also induced in a dose-dependent fashion. Baldivia et al. demonstrated the apoptotic effect of *Stryphnodendron adstringens* aqueous extract (SAAE) on B16F10Nex-2 melanoma cell line [35]. Likewise, SAAE at a dose of 500 µg/mL for 48 h had the highest apoptotic effect. Applying SAAE for 48 h resulted in the up-regulation of intracellular ROS level and c-caspase-3 activation in a concentration-dependent manner. Aimvijarn et al. claimed the induction of apoptosis by Thai water lily in B16 malignant melanoma cell line [36]. Thai water lily, applied at a dose of 200 to 1000 μg/mL for 24 h, resulted in apoptosis in a concentration-dependent manner. Apoptosis was induced through the stimulation of the transcription factor APF/Refl, which acted as an inhibitor of cellular invasion by acting as a cellular oxidant. Cicco et al. claimed the induction of apoptosis by cynaropicrin in A375 and SK-MEL-28 melanoma cell lines [30]. Cynaropicrin, applied at a dose of 30 μM for 24 and 48 h, induced apoptosis through the up-regulation of cleaved caspase-3 and cleaved caspase-7, and the down-regulation of Bcl-2 and NF-_K_B.

**Table 1 plants-14-00951-t001:** Apoptosis-inducing compounds.

Classification	Drug	Source	Cell Line/Animal Model	Dose/Duration	Efficacy	Mechanism	Reference
Animal	Porcupine bezoar	*Hystrix Brachyura*	A375	1, 2, 3, 4 µg/mL; 24, 48, 72, 96 h	Induction of apoptotic activity	↑Bax, cyt-c, c-caspase-3, c-caspase-9 ↓Bcl-2	[18]
Diarylheptanoid, Flavonoid	Curcumin, quercetin	*Curcuma longa* L.	A375	1.5, 3.1, 6.3, 12.5, 25, 50 μM; 24 h	Induction of apoptosis	↓DVL2, β-catenin, cyclin D1, Cox2, Axin2, Bcl-2 ↑c-caspase-3, c-caspase-7	[19]
Fungi	*Pestalotiopsis microspora*	*Gymnema sylvestre* leaves	B16F10	10, 20, 30, 40, 50, 100, 150, 200 μg/mL	Induction of apoptosis	↑ROS	[20]
Plant Extract	Aged garlic	*Allium sativum* L.	M14 WT	1, 5, 7, 10 mg/mL; 24, 48, 72 h	Initiation of intrinsic apoptosis	↑Cyt c, Smac, Diablo	[21]
Plant Extract	Cashew gum	*Anacardium occidentale* Linn	B16F10	3.125, 6.25, 12.5, 25, 50, 100 μg/mL; 72 h	Induction of cell death and apoptosis	↓γH2AX	[22]
			C57BL/6 mice	50, 100 mg/kg; 14 d			
Plant Extract	*C. decidua*, *C. edulis* *H. tuberculatum*,	*C. decidua,* *C. edulis* *H. tuberculatum*	A375, B16F10	0.45, 10, 15 µg/mL; 48 h	Induction of apoptosis	↑c-caspase-3, c-caspase-7	[23]
Plant Extract	Chamomile, parsley, celery	*Matricaria chamomilla* L., *Petroselinum crispum* (Mill.) Nym. ex A. W. Hill, *Apium graveolens* L.	A375	10, 30, 60 μg/mL; 72 h	Induction of apoptotic activity	↑c-caspase-3 ↓ IL-10	[24]
Plant Extract	*Conyza bonariensis* (L.)	*Conyza bonariensis* (L.)	SK-MEL-28	20 μg/mL; 48 h	Induction of apoptosis	↑ROS	[25]
Plant Extract	Curcumin, demethoxycurcumin, bisdemethoxycurcumin	*Curcuma longa* Linn	A2058	5, 10, 15, 20, 25 μΜ; 24, 48 h	Initiation of apoptosis		[26]
Plant Extract	Curzerene	*Eugenia uniflora*	HCT-116, AGP-01, SKMEL-19, MRC-5	5, 10 μM; 72 h	Induction of apoptosis		[27]
Plant Extract	Hibiscus, cinnamomum	*Hibiscus syriacus, Cinnamomum loureirii* Nees	G361	62.5, 125, 250, 500, 1000 μg/mL; 48 h	Induction of apoptosis		[28]
Plant Extract	*Leonotis nepetifolia*	*Leonotis nepetifolia*	A375	500 µg/mL; 24 h	Induction of apoptosis	↑p53	[29]
Plant Extract	Pine bark	*Pinus maritima*	A375	5, 25, 50, 100 µg/mL; 12, 24, 48 h	Induction of apoptosis	↑ROS, c-caspase-3 ↓MMP-9	[37]
Plant Extract	*Piptoporus betulinus*	*Piptoporus betulinus*	Hs27, A375, WM115	2.5, 5, 10 μL/mL; 24 h	Induction of apoptosis	↑c-caspase-3, c-caspase-7	[31]
Plant Extract	Portuguese propolis	*Apis mellifera* L.	A375, WM9	5, 10, 15, 20, 25, 30, 35, 40, 45, 50, 55, 60 µg/mL; 72 h	Activation of apoptosis	↑ROS, Bax, c-caspase-3, c-caspase-9, p53 ↓Bcl-XL, Bcl-2	[32]
Plant Extract	Red macroalgae	*Gelidium latifolium*	B16F10	10, 50, 100, 200 μg/mL; 72 h	Induction of apoptosis	↑p53, Bax, Bak ↓Bcl-2	[33]
Plant Extract	*Rhamnus alaternus*	*Rhamnus alaternus*	B16F10	10, 20, 40 µg/mL; 48 h	Induction of apoptosis		[34]
Plant Extract	*Stryphnodendron adstringens*	*Stryphnodendron adstringens*	B16F10	10, 25, 50, 125, 250, 500 µg/mL; 24, 48 h	Promotion of apoptosis	↑ROS, c-caspase-3	[35]
Plant Extract	Thai water lily	*Nymphaea stellate*	B16F10	200, 400, 600 800, 1000 μg/mL; 24 h	Induction of apoptosis		[36]
Terpenoid	Cynaropicrin	*Centaurea drabifolia subsp.detonsa*	A375	30 μM; 24, 48 h	Induction of apoptosis	↑c-caspase-3, c-caspase-7 ↓Bcl-2, NF-KB	[30]

Abbreviations: Bax, Bcl-2-associated X protein; cyt-c, cytochrome c; c-caspase, cleaved caspase; Bcl-2-antagonist/killer; DVL2, dishevelled segment polarity protein 2; β-catenin, beta-catenin; Cox2, prostaglandin–endoperoxide synthase 2; Axin2, axis inhibition protein 2; ROS, reactive oxygen species; Smac, second mitochondria-derived activator of caspase; Diablo, direct inhibitor of apoptosis-binding protein with low pI; γH2AX, phosphorylated histone H2AX; IL-10, interleukin 10; p53, cellular tumor antigen p53; MMP-9, matrix metalloproteinases 9; Bcl-XL, B-cell lymphoma—extra large.

### 3.2. Anti-Metastasis and Natural Products

Polyphenol-rich extract and terpenoid-type single compounds have exhibited anti-metastatic properties when exposed to malignant melanoma cells (Table 2 and Figure 3).

Xu et al. described anti-proliferation and anti-migration in melanoma-induced C57/BL6 mice [38]. Here, 30 mg/kg and 60 mg/kg of taxifolin, extracted from the roots of *Larix olgensis*, were intraperitoneally injected for 11 days. Proteins Ki67 and PCNA were down-regulated, resulting in the inhibition of metastasis. Silva et al. claimed the anti-metastatic property of *Eleutherine bulbosa* bulbs in melanoma-induced C57BL6 mice [39]. Mice treated with 100, 500, and 1000 μg/mL of plant extract for 11 days exhibited anti-metastatic effects. Thaichinda et al. reported the anti-metastatic properties of pine bark extract (PBE) in melanoma skin cancer cell migration and invasion [37]. A dose of 50 μg/mL of PBE for 45 h reduced the migration and invasion of A375 cells. Additionally, PBE downregulated MMP-9, a protein expressed by cell lines derived from advanced primary melanomas, thereby reducing cancer cell invasion. De Cicco et al. demonstrated the anti-metastatic effect of cynaropicrin [30]. The natural product inhibited cell proliferation, apoptosis, migration, and invasion, which decreased metastatic activity in A375 cells. The up-regulation of Nrf2 and down-regulation of phosho-ERK and p65 reduced the activation of ERK as well as the translocation of NF-_K_B into the nucleus, reducing the proliferation of human metastatic melanoma cells. Wu et al. reported hinokitiol’s anti-metastatic properties [40]. Hinokitiol showed an inhibitory effect on cell migration, leading to a decrease in metastatic activity in B16F10 cells. In C57BL/6 mice and BALB/c mice, injecting hinokitiol via the tail vein impeded tumor metastasis, which was shown through a decrease in the weight of the tumors. Proteins phospho-AKT and phospho-ERK were significantly suppressed, acting against cancer-promoting pathways.

### 3.3. Antioxidant and Natural Products

Single compounds ranging from flavonoids to glycosides, and extracts ranging from calcified material to herbaceous plants, possess antioxidant properties against malignant melanoma cells (Table 3 and Figure 4).

Prasedya et al. reported the antioxidative effect of *Sargassum cristaefolium (SCE)* on B16F10 melanoma cell [41]. At doses of 1, 10, 50, and 100 μg/mL for an hour, the cellular ROS level in cells exposed to SCE was significantly downregulated. Sunarwidhi et al. reported the antioxidant effects of Indonesian brown algae on the B16F10 malignant melanoma cell line [42]. Indonesian brown algae at doses of 50, 100, 500, 1000, and 4000 μg/mL for 16 h, evaluated by ABTS scavenging, resulted in augmented antioxidant activity in a dose-dependent manner. Antioxidant activity was here induced through the down-regulation of ROS. Khan et al. reported the antioxidant effects of Hystrix Brachyura Bezoar on the B375 malignant melanoma cell line [18]. Hystrix Brachyura Bezoar at doses of 1, 2, 3, and 4 μg/mL for 72 h, evaluated by dot–blot assays, resulted in antioxidant activity in a concentration-dependent manner. Antioxidant activity was induced through the down-regulation of ROS. Lee et al. claimed the antioxidant effects of *Lactobacillus gasseri* BNR17 on the B16-F10 melanoma cell line and HaCaT epidermal keratinocyte cell line [43]. *Lactobacillus gasseri* BNR17, at doses of 0.16, 0.312, 0.625, 1.25, 2.5, 5, and 10% *v*/*v* for 24 h, resulted in antioxidant activity in a concentration-dependent manner. Antioxidant activity was induced through the up-regulation of HO-1, CAT, GPX1 and SOD1 and the down-regulation of ROS. Kim et al. confirmed the antioxidative effects that the essential oils of *Chrysanthemum boreale MAKINO*(CBM) demonstrated in B16BL6 melanoma cells [44]. The antioxidant and anti-melanogenesis properties were evident when exposed to CBM at doses of 1, 10, and 30 μg/mL for 48 h. The effects were exhibited through the down-regulation of TYR and the up-regulation of p38, MAPK, and Erk1/2. Mustapha et al. documented the antioxidative trait of epicatechin extracted from Hawthorn, *Crataegus azarolus* L., on the melanoma cell line [45]. B16F10 cells were exposed to doses of 10, 20, 30, 40, and 50 μmol/L of epicatechin for 1 h. The ROS level diminished, exhibiting evidence of an antioxidative effect. Arung et al. illustrated the antioxidant effects of Glyasperin A on the B16 malignant melanoma cell line [46]. Glyasperin A at doses of 2.4, 4.8, 6.0, and 7.2 μM resulted in antioxidant activity in a concentration-dependent manner, determined by diphenyl-2-picrylhydrazyl (DPPH), 2,2′-azi- no-bis-3-ethylbenzthiazoline-6-sulphonic acid (ABTS), superoxide dismutase-like activity (SOSA), and oxygen radical absorbance capacity (ORAC) assays. Yang et al. claimed the cellular effects of *Pleurotus ferulae* in B16 melanoma cells [47]. Doses of 0.5, 1, 2, 5, and 10 μg/mL had antioxidant and antitumor activities by limiting cellular ROS activity. Cheimonidi et al. illustrated the induction of antioxidant activity in the B16F1 and B16F10 cell lines and C57BL/6 mice through exposure to acteoside [48]. Acteoside, at doses of 0.2, 0.4, 0.6, 0.8, and 1 mM for 24 h, resulted in the activation of related proteins in a dose-dependent manner. C57BL/6 mice were intraperitonially injected with acteoside at doses of 1 and 2.5 mg/mouse for 13 consecutive days. Antioxidant activity was induced through the up-regulation of Akt, CREB1, MEK1 and WNK1 and the down-regulation of STAT1 and STAT3. Lee et al. demonstrated the antioxidant effect of *Hygrophila erecta* [49]. The natural product induced the antioxidant defense system, apoptotic effects, inflammatory responses, and a melanin synthesizing effect in both NHDF and B16F1 cell lines. *Hygrophila erecta* at doses at doses of 1, 2, 4, 8, 16, 31, 62, 125, 250, 500, and 1000 μg/mL for 24 h resulted in the suppression of ROS levels and NO concentration associated with the activation of SOD and Nrf2. Ohkubo et al. reported the antioxidant effects of aged garlic extract (AGE) on the human malignant melanoma M14 wild type cell line [21]. AGE extract at doses of 1, 5, and 10 mg/mL for 40 min induced oxidative stress by decreasing mitochondrial sulfhydryl groups in a dose-dependent manner. Antioxidant activity was induced through the down-regulation of ROS. Kaushik et al. demonstrated the antioxidant effects of *Cinnamomum loureirii* Nees on the G361 malignant melanoma cell line [28]. *Cinnamomum loureirii* Nees at a dose of 1 mg/mL for 30 min, evaluated by DPPH free radical inhibition, resulted in augmented antioxidant activity in a dose-dependent manner. Antioxidant activity was induced through the down-regulation of ROS. Žitek et al. demonstrated the antioxidant effects of *Curcuma longa*, *Lycium barbarum*, *Equisetum arvense*, *Vitis vinifera*, and *Rosmarinus officinalis* extracts on the WM-266-4 metastatic melanoma cell line [50]. The extracts from the raw materials, applied at doses of 0.001, 0.01, 0.1, 1, 5, 10, 20, and 100 μg/mL for 24 h, resulted in antioxidant activity in a concentration-dependent manner, mediated through the down-regulation of ROS. Also, Žitek et al. claimed the antioxidant effects of ginger and hemp on the B16F10 malignant melanoma cell line [51]. A combination of hemp extract and ginger, at doses of 0.01, 0.1, 1, 5, 10, 20, and 50 mg/mL for 24 h, exhibited antioxidant activity in a concentration-dependent manner by down-regulating ROS levels. Chatatkun et al. claimed the antioxidant effects of *Garcinia atroviridis* on the B16F10 malignant melanoma cell line [52] at doses of 62.5, 125, 250, 500, and 1000 μg/mL, in a dose-dependent manner, through free radical scavenging activity, as determined by DPPH and ABTS assays. Antioxidant activity was induced through the down-regulation of ROS. Spera et al. demonstrated the antioxidant effects of *Hymenaea courbaril* L. on the B16F10 malignant melanoma cell line [53]. *Hymenaea courbaril* L., at a dose of 250 μg/mL for 30 min, evaluated by DPPH radical scavenging activity, resulted in antioxidant activity in a dose-dependent manner. Antioxidant activity was induced through the down-regulation of ROS. Díaz et al. claimed the antioxidant effects of *Laminaria ochroleuca*, *Porphyra umbilicalis* and *Gelidium corneum* on G-361 malignant melanoma cell line [54]. At doses of 25, 50, 75, 100, 150, 200, 300, 400, and 500 μg/mL for 16 h, these extracts demonstrated antioxidant activity in a dose-dependent manner through the down-regulation of ROS. Raja et al. demonstrated the antioxidant effects of *Lawsonia inermis* L. on B16F10 malignant melanoma cell line-bearing C57BL/6 mice [55]. *Lawsonia inermis* L. at doses of 500 mg/kg and 1000 mg/kg for 40 days resulted in antioxidant activity in a concentration-dependent manner. Antioxidant activity was induced through the down-regulation of GSH-Px, SOD, and CAT. Merecz-Sadowska et al. demonstrated the antioxidant effects of *Leonotis nepetifolia* (L.) R. Br. extract on the A375 malignant melanoma cell line [29]. *Leonotis nepetifolia* (L.) R. Br., applied at a dose of 10 μg/mL for 16 h, evaluated by ABTS scavenging, resulted in increased antioxidant activity in a dose-dependent manner. Antioxidant activity was induced through the down-regulation of ROS and Mdm-2. Furthermore, Quassinti et al. claimed the antioxidant effects of *Liriodendron tulipifera* L. on the A375 malignant melanoma cell line [56]. *Liriodendron tulipifera* L., at a dose of 5 mg/mL for 72 h, evaluated by DPPH, ABTS, and ferric reducing antioxidant power (FRAP) assays, resulted in antioxidant activity in a concentration-dependent manner. Antioxidant activity was induced through the down-regulation of ROS. Sipos et al. reported the antioxidant effects of *Melissa officinalis* L. on HaCaT human keratinocyte and A375 malignant melanoma cell line [57]. *Melissa officinalis* L., at doses of 0.1, 0.5, 1, 3, and 5 mg/mL for 20 min, evaluated by DPPH free radical consumption, resulted in enhanced antioxidant activity in a concentration-dependent manner through the down-regulation of ROS. Akaberi et al. showed the antioxidative property of *Nepeta sintenisii* Bornm [58]. B16F10 melanoma cells treated with 50 μg/mL of *Nepeta sintenisii* Bornm showed the suppression of oxidative stress and TYR in the cell line. Danciu et al. illustrated the antioxidant effect of parsley on the A375 melanoma cell line [24]. Dosages of parsley ranging from 10 to 60 μg/mL exposed to a melanoma cell line for 72 h resulted in the activation of various proteins in a dose-dependent manner. Antioxidant activity was induced by up-regulating c-casapse-3 and down-regulating IL-10. Wikiera et al. reported the effects of pectin extracted from dried apple pomace [59]. An increased extracellular ROS level was observed through 24-h exposure to 0.2, 0.4, 0.6, 0.8, and 1 mg/mL, encouraging antioxidant activities in B16F10 cells. Jeong et al. reported the antioxidant effects of *Penthorum chinense* Pursh on the B16-F10 melanoma cell line and HaCaT epidermal keratinocyte cell line [60]. *Penthorum chinense* Pursh at doses of 50 and 100 μg/mL for 24 h resulted in antioxidant activity in a concentration-dependent manner. Antioxidant activity was induced through the down-regulation of ROS, IL-6, COX-2, p-ERK, p-p38, p-JNK, and MMP. Nanni et al. claimed the antioxidant effect of *Spartium junceum* L. on the B16F10 malignant melanoma cell line [61]. *Spartium junceum* L., at doses of 4, 8 and 10 mg/mL for 4, 24, and 48 h, inhibited the growth of the melanoma cell line, which resulted in the activation of various proteins in a dose- and time-dependent manner. Antioxidant activity was induced through the down-regulation of MITF. Baldivia et al., demonstrating the antioxidant effect of *Stryphnodendron adstringens* aqueous extract (SAAE) [35]. SAAE at a dose of 50 µg/mL had the highest antioxidant effect on B16F10Nex-2 melanoma cell line. SAAE showed antioxidant activity by scavenging ABTS and DPPH free-radicals, and proved its antioxidant capacity through reduced malondialdehyde (MDA) levels. Aimvijarn et al. claimed the antioxidant effects of Nymphaea stellate extracted from Thai water lily on the B16F10 malignant melanoma cell line [36]. Nymphaea stellate, applied at doses of 200, 400, 600 800, and 1000 μg/mL for 24 h, evaluated by fluorescent 2,7-dichlorodyhydrofluorescein diacetate (DCFH-DA), resulted in antioxidant activity in a concentration-dependent manner. Antioxidant activity was induced through the down-regulation of ROS. Ciorîţă et al. reported the induction of antioxidant activity by *Vinca minor, V. herbacea*, *V. major* and *V. major var. variegata* against HaCaT and the A375 malignant melanoma cell line [62]. *Vinca minor*, *V. herbacea*, *V. major* and *V. major var. variegata* extracts, at concentrations of 0.09, 0.5, 1, 2, and 3% for 24 h and 72 h, resulted in antioxidant activity in a dose-dependent manner, along with antibacterial and antitumor effects. Zhong et al. claimed the induction of antioxidant activity against the B16F10 cell line and C57BL/6 mice through exposure to 2′,3,4,4′-tetrahydrochalcone [63]. 2′,3,4,4′-tetrahydrochalcone, at doses of 1, 10, 20, and 40 μM/L for 30 min, resulted in the activation of related proteins in a dose-dependent manner. C57BL/6 mice were injected with 2′,3,4,4′-tetrahydrochalcone in doses of 8.5 and 10 μg for 40 days. Antioxidant activity was induced through the down-regulation of ROS. Feng et al. reported the antioxidant effects of nobiletin on the SK-MEL-28 malignant melanoma cell line [64]. Nobiletin, applied at doses of 5, 15, and 45 μM for 30 min and evaluated by DCFH-DA, where the collected cells’ fluorescence intensity was detected using a flow cytometer, resulted in increased antioxidant activity in a dose-dependent manner. Antioxidant activity was induced through the up-regulation of Keap1 and down-regulation of Nrf2 and ROS. Cicco et al. claimed the induction of antioxidant activity of cynaropicrin on A375 melanoma cell lines [30]. Cynaropicrin, applied at doses of 3, 10, and 30 μM for 24 h and 48 h, resulted in the activation of antioxidant activity in a concentration-dependent manner. Antioxidant activity was induced through the up-regulation Nrf2 and down-regulation of pERK and p65. Mokhtari et al. illustrated the antioxidant effects of maslinic acid on A10 and B16F10 malignant melanoma cell lines [65]. Maslinic acid, applied at doses of 10.6, 21.6, 42.3, and 84.6 μM for 24 h, resulted in the activation of antioxidant activity in a dose-dependent manner. Antioxidant activity was induced through the down-regulation of ROS, CAT, G6PDH, SOD, GST, GPX, and GR. Alvarado et al. demonstrated the antioxidant effects of oleanolic acid and ursolic acid extract from *Plumeria obtusa* on B16 malignant melanoma cell line [66]. Oleanolic acid and ursolic acid at a dose of 2 mg/mL for 90 min, evaluated by DPPH free radical activity, resulted in antioxidant activity in a dose-dependent manner. Antioxidant activity was induced through the down-regulation of ROS.

### 3.4. Anti-Angiogenesis and Natural Product

Extracts from natural compounds that have already been addressed possess properties of anti-angiogenesis on malignant melanoma cells (Table 4).

Sipos et al. claimed the induction of angiogenesis in HaCaT human keratinocyte, A375 malignant melanoma cell line and SKH-1 mice through exposure to *Melissa officinalis* L. extract [57]. *Melissa officinalis* L., at doses of 20, 100, 250, 500, and 1000 μg/mL for 24 h, resulted in the activation of proteins in a concentration-dependent manner. SKH-1 mice were topically applied with *Melissa officinalis* L. at a dose of 5 mg/mL every 2 days for 2 weeks. This resulted in the angiogenetic behavior of the specified melanoma cell.

## 4. Discussion

Cancer is related to an imbalance between reactive oxygen species (ROS) and antioxidants, as well as increased oxidative stress. Metabolic disturbances and signaling abnormalities cause the ROS level to increase, and eventually result in carcinogenesis [67]. Apoptosis is a form of programmed cell death. Cells that malfunction due to various changes in the body activate self-cell death [68]. However, most of the cells that do not notice their defect fail to turn on self-cell death systems, and form cancer. All cancers have the inclination to metastasize, which is when cancer cells break away from the primary location and move to secondary and tertiary locations [69]. The cancer cells go through this process by traveling through lymph nodes or blood vessels, which is angiogenesis [70]. Oxidative stress also plays a critical role in melanoma progression, as excessive reactive oxygen species (ROS) can promote DNA damage and alter key signaling pathways [71]. Curcumin has shown potential use as an adjuvant therapy in cancer treatment. A clinical trial combining curcumin with gemcitabine in pancreatic cancer reported a partial response and stable disease in some patients, highlighting its possible role in melanoma [72]. Figure 5 illustrates the chemical structures of key natural compounds discussed in this review, including curcumin (PubChem CID: 969516), quercetin (PubChem CID: 5280343), acteoside (PubChem CID: 5281800), cynaropicrin (PubChem CID: 119093), and hinokitiol (PubChem CID: 3611). These compounds have been reported to exhibit significant biological activities in melanoma treatment, such as apoptosis induction, oxidative stress regulation, and anti-metastatic effects. According to the Skin Cancer Foundation, a form of cancer that begins in a type of cell known as a melanocyte is melanoma. Among various skin cancers, like basal cell carcinoma and squamous cell carcinoma, malignant melanoma is the most dangerous type of skin cancer due to its ability to metastasize to other organs more rapidly if it is not treated at an early stage [73]. Various types of treatment methods that are available for curing melanoma have been approved by the FDA. Yet, due to its extreme tendency to relapse, melanomas do not always respond to these therapeutics [74]. Also, the death of melanoma cells is insignificant even with chemotherapy, and the treatment itself reduces the patients’ quality of life. Chemotherapy causes vomiting, nausea, fever, perforations, severe infection from within, diarrhea, constipation, and allergic reactions [75]. Thus, this calls for the adjuvant treatment of existing cancer therapeutics with natural products in curing melanoma. Natural products will provide patients with an easier approach to cure melanoma, one of the most commonly occurring cancers that roots from the endocrine systems in the body.

In this systematic review, the antioxidant activity and anti-cancer activity of natural products against melanoma were scrutinized. A375 and SK-MEL-28 are human melanoma cell lines that are widely used to assess proliferation and apoptotic responses to natural compounds [76]. B16F10, a murine melanoma cell line, is commonly employed in metastatic models due to its high invasiveness [77]. WM115, derived from primary melanoma, is utilized in studies investigating early-stage melanoma progression and genetic variations [78]. Understanding these distinctions is essential for accurately interpreting in vitro findings and evaluating the therapeutic potential of natural compounds in melanoma. Furthermore, the classification, origin, experimental model, dose, duration, efficacy, and mechanisms of action cited in studies covering single compounds up to complex decoctions were systematically categorized. In total, four tables were assembled based on the anti-cancer mechanisms of the natural products. The systematic reviews published within the last 10 years regarding the effects of natural products on melanoma were mainly included. This comprehensive approach allows for a clear understanding of the potential of various natural products. In addition, the detailed organization of the data in each of the tables facilitates the understanding and potential replication of our experimental results. The existing reviews have focused more on the effects of natural products on the adverse symptoms caused by adjuvant treatments. The antioxidant activity and anti-cancer activity of all the natural products included in this review were selective to melanoma cancer cell lines. Among several systematic reviews regarding melanoma, Garbe et al. only included therapeutics in clinical practice, and Al Qathama et al. focused only on natural active principles. There were no systematic reviews that solely examined the association between melanoma and natural products in general. However, in this paper, all studies that discussed the intrinsic efficacies of natural products were analyzed by forming four different categories of cancer mechanisms. In particular, in some studies, *Cinnamomum loureirii* Nees, which down-regulates ROS, acetoside, which up-regulates Akt, CREB1, MEK1, and WNK1 and downer-gulates STAT1 and STAT3, and pectin, which down-regulates ROS, were administrated at a low concentration of 1 mg/mL or less, illustrating their potential to be utilized as main compounds for treating melanoma.

However, there are several drawbacks to this systematic review. First, it only evaluated studies published in English, which may have resulted in a lack of comprehensive data. Furthermore, the lack of clinical trials and the shortage of research on anti-angiogenesis and anti-metastasis effects related to natural products and melanoma are problems. Future studies should include additional search engines beyond databases such as Google Scholar and PubMed, so as to gather more extensive data, including studies published in other languages. While twenty studies on apoptosis and thirty-four studies on antioxidant effects were identified, there was a limited number of studies related to anti-angiogenesis and anti-metastasis, requiring the further exploration of these categories. Moreover, some of the studies lacked either in vitro or in vivo investigations, highlighting the need for further examinations to exemplify the effects of natural products on melanoma. Studies focusing on *Stryphnodendron adstringens*, Glyasperin A, *Eleutherine bulbosa bulbs*, *Rhamnus alaternus*, Thai water lily, *Hibiscus* and *Cinnamomum*, Curzerene, and *Curcuma longa* Linn also require additional in-depth mechanistic research in order to better understand the impacts of natural products on melanoma. It is essential to elucidate how these substances work at the cellular level, as optimizing the efficacy of natural product-based therapies depends on understanding their molecular mechanisms. Considering these factors, broader studies on the effects of natural substances on melanoma are strongly advised. Despite its limitations, this systematic review offers an overview of the potential therapeutic benefits of natural products in treating melanoma, and highlights their significant role in cancer therapy.

## 5. Conclusions

This study has reviewed the anti-cancer effects of natural products on melanoma by classifying forty-three studies in terms of antioxidant and anti-cancer mechanisms. The regulatory factors that activate anti-cancer activity were presented for an overview of the effects of natural products. Natural products are promising, as they have the capacity to treat melanoma. Therefore, additional in vitro, in vivo and clinical studies should be conducted in order to fully utilize and develop such prospective resources.

## Figures and Tables

**Figure 1 plants-14-00951-f001:**
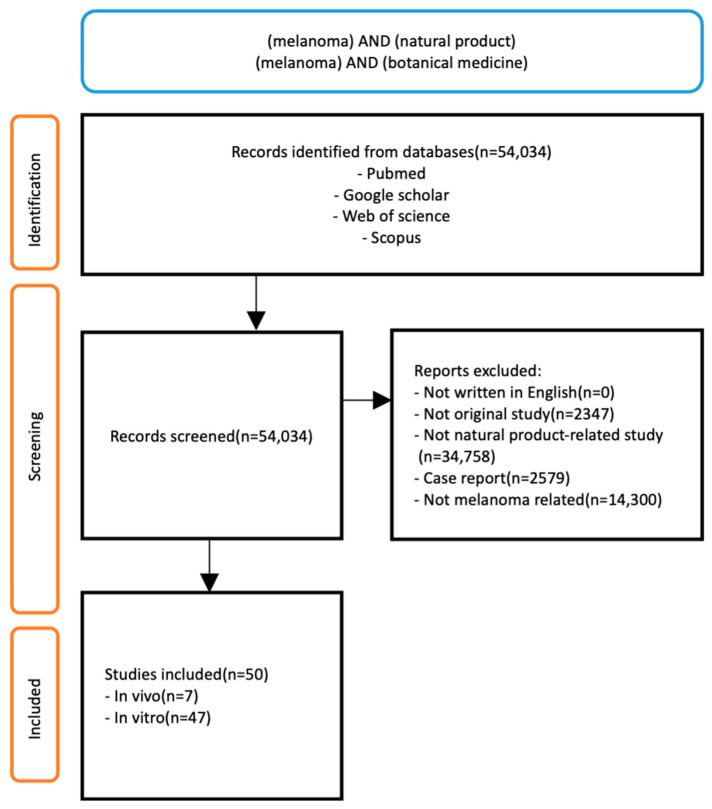
Flowchart of included studies.

**Figure 2 plants-14-00951-f002:**
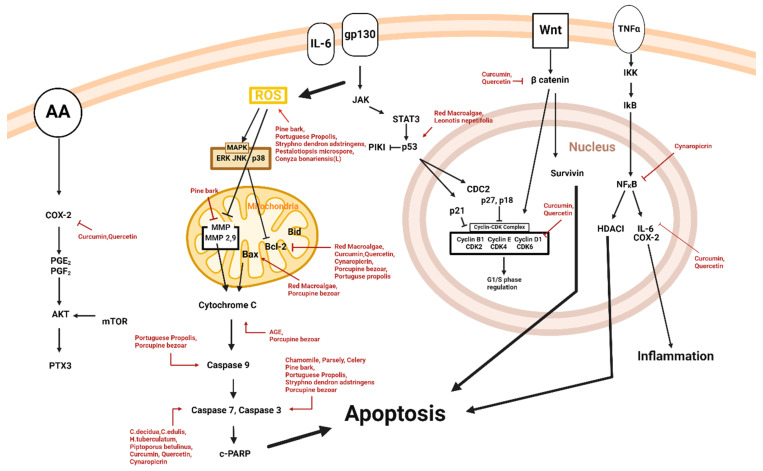
Schematic diagram of apoptotic mechanisms of natural products.

**Figure 3 plants-14-00951-f003:**
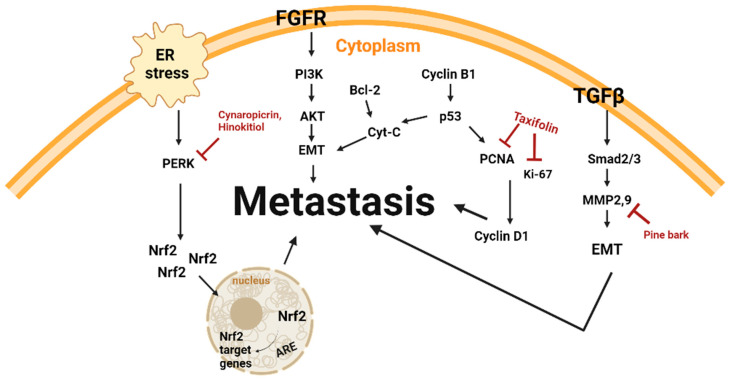
Schematic diagram of metastasis mechanisms and natural products.

**Figure 4 plants-14-00951-f004:**
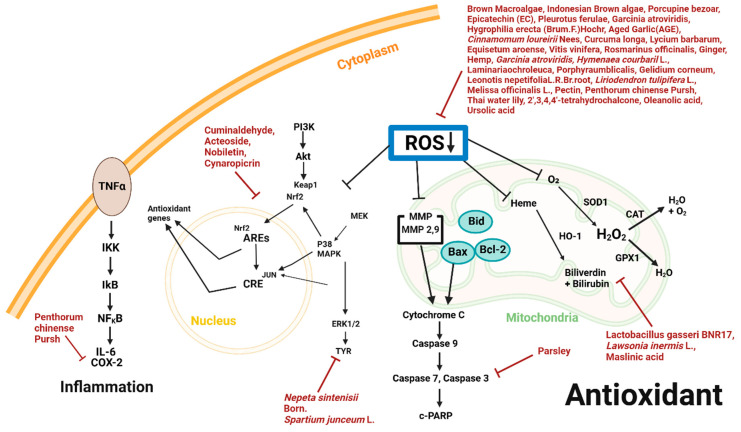
Schematic diagram of antioxidant mechanisms and natural products.

**Figure 5 plants-14-00951-f005:**
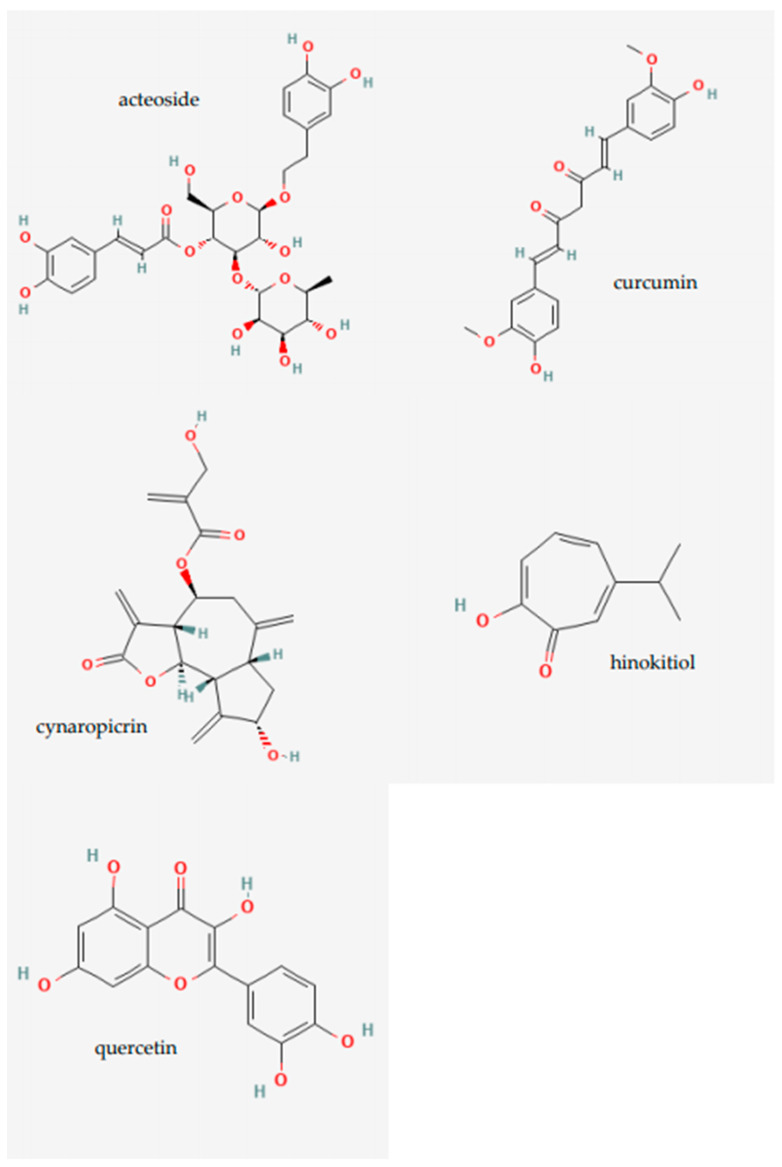
Chemical structures of selected natural compounds with promising anti-melanoma activity: acteoside (PubChem CID: 5281800), curcumin (PubChem CID: 969516), cynaropicrin (PubChem CID: 119093), hinokitiol (PubChem CID: 3611), and quercetin (PubChem CID: 5280343).

**Table 2 plants-14-00951-t002:** Anti-metastasis-inducing compounds.

Classification	Compound	Source	Cell Line/Animal Model	Dose/Duration	Efficacy	Mechanism	Reference
Flavonoid	Taxifolin	*Larix olgensis* roots	C57/BL6 mice	30, 60 mg/kg; 11 d	Inhibition of metastasis and proliferation	↓PCNA	[38]
Plant Extract	*Eleutherine bulbosa* bulbs	*Eleutherine bulbosa*	C57BL6 mice	100, 500, 1000 μg/mL; 14 d	Anti-metastatic property		[39]
Plant Extract	Pine bark	*Pinus maritima*	A375	1, 5, 25, 50 μg/mL; 45 h	Inhibition of metastasis	↓MMP-9	[37]
Terpenoid	Cynaropicrin	*Centaurea drabifolia* subsp. *detonsa*	A375	3, 10 μM; 0, 24, 48 h	Inhibition of metastasis	↑Nrf2 ↓pERK, p65	[30]
Terpenoid	Hinokitiol	*Chamaecyparis taiwanensis*	B16F10	1250 nM; 0, 6, 12 h	Inhibition of metastasis	↓pAKT, pERK	[40]
			C57BL/6 mice, BALB/c mice	1250 nM; 15 d			

Abbreviations: PCNA, proliferating cell nuclear antigen; MMP-9, matrix metallopeptidase; Nrf2, nuclear factor erythroid-2-related factor 2; pERK, phospho-extracellular-regulated kinase; p65, nuclear factor NF-kappa-B p65 subunit; pAKT, phospho-protein kinase B.

**Table 3 plants-14-00951-t003:** Antioxidant property-inducing single compounds.

Classification	Compound	Source	Cell Line/Animal Model	Dose/Duration	Efficacy	Mechanism	Reference
AlgaeProtista	Brown macroalgae	*Sargassum cristaefolium (SCE)*	B16F10	1, 10, 50, 100 μg/mL; 1 h	Initiation of antioxidant properties	↓ROS	[41]
AlgaeProtista	Indonesian Brown algae	*Sargassum polycystum, Sargassum cristaefolium, Sargassum aquifolium Turbinaria ornata*	B16F10	50, 100, 500, 1000, 4000 μg/mL; 16 h	Induction of antioxidant activity	↓ROS	[42]
Animal	Porcupine bezoar	*Hystrix Brachyura* Bezoar	A375	1, 2, 3, 4 μg/mL; 72 h	Induction of antioxidant capacity	↓ROS	[18]
Bacteria	Lactobacillus gasseri BNR17		B16-F10, HaCaT	0.16, 0.312, 0.625, 1.25, 2.5, 5, 10% *v*/*v*	Induction of antioxidant activity	↑HO-1, CAT, GPX1, SOD1 ↓ROS	[43]
Benzaldehyde	Cuminaldehyde	*Chrysanthemum boreale* MAKINO	B16BL6	1, 10, 30 μg/mL; 48 h	Induction of antioxidant response and antimelanogenesis properties	↑Akt, CREB1, MEK1, WNK1, TAT1, STAT3, p38 MAPK, Erk1/2 ↓TYR	[44]
Flavonoid	Epicatechin (EC)	*Crataegus azarolus* L.	B16F10	10, 20, 30, 40, 50 μmol/l; 1 h	Antioxidant property	↓ROS	[45]
Flavonoid	Glyasperin A	*Macaranga pruinosa*	B16	2.4, 4.8, 6.0, 7.2 μM	Induction of antioxidant activity		[46]
Fungi	*Pleurotus ferulae*	*Pleurotus ferulae*	B16, Eca-109, BGC823, HeLa	0.5, 1, 2, 5, 10 mg/mL	Induction of antioxidant, antitumor activities	↓ROS	[47]
Phenylethanoid	Acteoside	*Lippia citriodora*	B16F1, B16F10	0.2, 0.4, 0.6, 0.8, 1 mM; 24 h	Induction of antioxidant activity	↑Akt, CREB1, MEK1, WNK1 ↓ ROS, STAT1, STAT3	[48]
Plant Extract	4-methoxycinnamic acid, 4-methoxybenzoic acid, methyl linoleate, asterriquinone C-1	*Hygrophila erecta* (Brum. F.) Hochr.	B16F10	1, 2, 4, 8, 16, 31, 62, 125, 250, 500, 1000 μg/mL; 24 h	Induction of antioxidative role	↑SOD, Nrf2 ↓ROS, NO	[49]
Plant Extract	Aged garlic extract (AGE)	*Allium sativum* L.	M14 WT	1, 5, 10 mg/mL; 40 min	Induction of antioxidant activity	↓ROS	[21]
Plant Extract	*Cinnamomum loureirii* Nees	*Cinnamomum loureirii* Nees	G361	1 mg/mL; 30 min	Induction of antioxidant activity	↓ROS	[28]
Plant Extract	*Curcuma longa, Lycium barbarum, Equisetum arvense, Vitis vinifera, Rosmarinus officinalis*	*Curcuma longa, Lycium barbarum, Equisetum arvense, Vitis vinifera, Rosmarinus officinalis*	WM-266-4	0.001, 0.01, 0.1, 1, 5, 10, 20, 100 μg/mL; 24 h	Induction of antioxidant activity	↓ROS	[50]
Plant Extract	Ginger, hemp	*Zingiber officinale, Cannabis sativa* L.	B16F10	0.01, 0.1, 1, 5, 10, 20, 50 mg/mL; 24 h	Induction of antioxidant activity	↓ROS	[51]
Plant Extract	*Garcinia atroviridis*	*Garcinia atroviridis* Griff. ex. T. Anderson	B16F10	62.5, 125, 250, 500, 1000 μg/mL	Induction of pro-antioxidant activity	↓ROS	[52]
Plant Extract	*Hymenaea courbaril* L.	*Hymenaea courbaril* L.	B16F10	250 μg/mL	Induction of antioxidant activity	↓ROS	[53]
Plant Extract	*Laminaria ochroleuca, Porphyra umbilicalis, Gelidium corneum*	*Laminaria ochroleuca, Porphyra umbilicalis, Gelidium corneum*	G-361	25, 50, 75, 100, 150, 200, 300, 400, and 500 μg/mL; 16 h	Induction of antioxidant property	↓ROS	[54]
Plant Extract	*Lawsonia inermis* L.	*Lawsonia inermis* L.	C57BL/6 mice	500, 1000 mg/kg; 40 d	Exhibition of optimum antioxidant activity and protection against oxidative stress	↓GSH-Px, SOD, CAT	[55]
Plant Extract	*Leonotis nepetifolia* (L.) R. Br. root	*Leonotis nepetifolia* (L.) R. Br.	A375	10 μg/mL; 16 h	Possession of antioxidant activity	↓ROS, Mdm-2	[29]
Plant Extract	*Liriodendron tulipifera* L.	*Liriodendron tulipifera* L.	A375	5 mg/mL; 72 h	Induction of antioxidant activity	↓ROS	[56]
Plant Extract	*Melissa officinalis* L.	*Melissa officinalis* L.	HaCaT, A375	0.1, 0.5, 1, 3, 5 mg/mL; 20 min	Induction of antioxidant activity	↓ROS	[57]
Plant Extract	*Nepeta sintenisii* Born.	*Nepeta sintenisii* Born.	B16F10	50 μg/mL; 24 h	Induction of antioxidant activity	↓ROS, TYR	[58]
Plant Extract	Parsley	Parsley	A375	10, 30, 60 μg/mL; 72 h	Induction of antioxidant and anti-inflammatory activity	↑c-caspase-3 ↓ IL-10	[24]
Plant Extract	Pectin	Dried apple pomace	HT-29, B16F10	0.2, 0.4, 0.6, 0.8, 1 mg/mL; 24 h	Induction of antioxidant activity	↓ROS	[59]
Plant Extract	*Penthorum chinense* Pursh	*Penthorum chinense* Pursh	B16-F10, HaCaT	50, 100 μg/mL; 24 h	Induction of antioxidative effect	↓ROS, IL-6, COX-2, p-ERK, p-p38, p-JNK, MMP	[60]
Plant Extract	*Spartium junceum* L.	*Spartium junceum* L.	B16F10	4, 8, 10 mg/mL; 4, 24, 48 h	Induction of antioxidant activity	↓ MIFT	[61]
Plant Extract	*Stryphnodendron adstringens* (Mart.) Coville (Fabaceae)	*Stryphnodendron adstringens* (Mart.) Coville (Fabaceae)	B16F10Nex-2	10, 50 μg/mL;	Induction of antioxidant activity		[35]
Plant Extract	Thai water lily	*Nymphaea stellate*	B16F10	200, 400, 600, 800, 1000 μg/mL; 24 h	Induction of antioxidant activities	↓ROS	[36]
Plant Extract	*Vinca minor,* *V. herbacea,* *V. major,* *V. major* var. *variegata*	*Vinca minor,* *V. herbacea,* *V. major,* *V. major* var. *variegata*	HaCaT, A375	0.09, 0.5, 1, 2, 3%; 24, 72 h	Induction of antioxidant, antibacterial and antitumor activity		[62]
Polyphenol	2′,3,4,4′-tetrahydrochalcone	*Vernohia anthelmintica* (L.) *willd*	B16F10	1, 10, 20, 40 μM/L; 30 min	Induction of antioxidant activity	↓ROS	[63]
			C57BL/6 mice	8.5, 10 μg; 40 d			
Polyphenol	Nobiletin	*Citrus unshiu Markovich*	SK-MEL-28	5, 15, 45 μM; 30 min	Induction of antioxidant activity	↑Keap1 ↓Nrf2, ROS	[64]
Terpenoid	Cynaropicrin	*Centaurea drabifolia* subsp. *detonsa*	A375	3, 10 μM; 0, 24, 48 h	Inhibition of metastasis	↑Nrf2 ↓pERK, p65	[30]
Triterpenoid	Maslinic acid	*Olea europaea* L.	A10, B16F10	10.6, 21.6, 42.3, 84.6 μM; 24 h	Induction of antioxidant activity	↓ROS, CAT, G6PDH, SOD, GST, GPX, GR	[65]
Triterpenoid	Oleanolic acid, Ursolic acid	*Plumeria obtusa*	B16	2 mg/mL; 90 min	Induction of antioxidant activity	↓ROS	[66]

Abbreviations: ROS, reactive oxygen species; HO-1, heme oxygenase 1; CAT, catalase; GPX, glutathione peroxidase; SOD, superoxide dismutase; Akt, protein kinase B; CREB, cAMP responsive element binding protein; MEK, mitogen-activated protein kinase; WNK, lysine deficient protein kinase; TAT, trans-activator of transcription; STAT, signal transducer and activator of transcription; p38 MAPK, p38 mitogen-activated protein kinase; Erk1/2, extracellular-regulated kinase 1/2; TYR, tyrosinase; Nrf2, nuclear factor erythroid-2-related factor 2; NO, nitric oxide; GSH-Px, glutathione peroxidase 1; Mdm-2, mouse double minute 2 homolog; c-caspase, cleaved caspase; IL, interleukin; COX-2, prostaglandin-endoperoxide synthase 2; p-ERK, phospho-extracellular-regulated kinase; p-p38, phosphorylated p38 mitogen-activated protein kinases; p-JNK, phosphorylated stress-activated protein kinase; MMP, matrix metallopeptidase; MITF, microphthalmia-associated transcription factor; Keap1, kelch-like ECH-associated protein 1; p65, nuclear factor NF-kappa-B p65 subunit; G6PDH, glucose-6-phosphate 1-dehydrogenase; GST, glutathione S-transferase; GPX, glutathione peroxidase; GR, glutathione reductase.

**Table 4 plants-14-00951-t004:** Anti-angiogenesis-inducing single compounds.

Classification	Compound	Source	Cell Line/Animal Model	Dose/Duration	Efficacy	Mechanism	Reference
Plant Extract	*Melissa officinalis* L.	*Melissa officinalis* L.	HaCaT, A375	20, 100, 250, 500, 1000 μg/mL; 24 h	Induction of antiangiogenic and antioxidant activity		[57]
			SKH-1 mice	5 mg/mL; 14 d			

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
