# Peer review of "Natural Products for Melanoma Therapy: From Traditional Medicine to Modern Drug Discovery"

_plants, 2025, doi:10.3390/plants14060951_

Round 1

Reviewer 1 Report

Comments and Suggestions for Authors

Dear Authors,

I have reviewed the manuscript and have come to the following conclusions.

The topic of the manuscript is the treatment of melanoma with herbal extracts and herbal remedies. The article is a comprehensive review.

The topic is not general, but may find a place in the journal, with some modifications.

The manuscript would attract many readers as many people are sent with this disease.

My observations are as follows:

Basically, the article is not properly formatted with cross references, which is a basic requirement of the journal and MDPI. Why is this the case, why did the authors not prepare this?

Furthermore, the figures and tables are in order and should be.

The literature used is correct.

I suggest a major revision that covers the whole manuscript.

The text is not coherent. The ideas are not connected at all, each publication is in a new paragraph - this causes the text to fall apart completely. Therefore, the whole text needs to be rewritten, the text needs to be flowing. The manuscript is focused on the melanoma and the plants take a back seat. This should be reversed by rewriting the text. The plants should be in the foreground, that should be the main point, and the melanoma should be next to it, in a naturalistic way, but not so strongly. I ask for the amendments. There should be some structuring, because there is no skeleton in the text, although the subject is very good.

Reviewer 2 Report

Comments and Suggestions for Authors

Authors present a systematic review entitled "Natural products for melanoma therapy: From Traditional Medicine to Modern Drug Discovery". As the sources of origincal studies are well explained and well explained, authors could perform some modifications to upgrade the quality of the manuscript. 

Introduction section

Q1. Authors could reoganize this section to better understand the review's context. Authors could explain first the global burden of melanoma globally, then in Asia (or in others geographical zones) and finally discuss the current treatment options. 

Q2. Authors must better emphasize the importantce of natural products in the treatment of melanoma in this section. They mention the interest for adjuvant treatment and synergistic effect, but mostly studies mentionned in the review report the efficacy against melanoma in vitro by single components (e.g. extract or pure natural compound). If studies between natural products/anti-cancer drugs are not available, authors must modify the introduction section to underline in this case the importance of natural products as source of bioactive compounds to treat melanoma. 

Q3. Rephrase: This points to one of the crucial takeaways and prompt of this research, which is that melanoma mostly affected and still is affecting those in Europe and North America, but it is slowly showing its prevalence in Asia in the recent years.

Q4. Add ref:

melanoma affected 20 million people around the globe in 2022.

Fortunately, when detected early, melanoma can be effectively treated.

and early detection could lead to a successful treatment.

Q5. Authors mention in line 58 that melanoma is prelevent in people in their mid-40 and in the line 73, in the population over 50. Please verify. 

Q6. In line 80, authors could add the percentage of successuful treatment reported for melanoma. 

Results Section

Q7. In general, authors could include chemical composition information (if available) for extracts mentionned in this section.

Q8. Tables 1 and 2. Please, modify the title as not only pure natural products are indicated. Authors can add extract preparation information in "Compound" column in Tables. 

Q9. Authors mention as "natural products" plant/microorganisms extracts. Please, modify accordantly (e.g. line 129, authors mention as natural product the porcupine bezoar extract). 

Q10. In lines 133 and 165, authors present two studies about curcumin. These studies present similar results/degree of activity ? Authors must underline if the studies report coherent results. 

Q11. For the study mentioned in lines 163-166, are there any SAR elements to explain for these derivatives ? 

Q12. In table 1, scientific names must be in italic. 

Q13. In lines 188/189, please verify the author name. 

Q14. For the study treated in lines 272/276, authors must indicate if these findings are in accordance with the results of the same extracts mentioned in lines 127/131. Same for the case of Stryphnodendron in line 390 and 204. Even if are the same publications, authors could link both kind of activities. 

Discussion section 

Q15. Authors could modify this section to highlight the difference of each activity treated in the review (apoptosis vs anti-metastasis vs antioxidant vs antiagenesis) and the potential of natural products as adjuvants to treat melanoma cancers. Are there any cases of natural adjuvant to treat melanoma or another cancers ? Clinical trials ? 

Q16. Authors mention in line 503 that acetoside could be used to treat melanoma. There are any in vivo study to support this point ? 

Q17. Plaese check police size in line 532. 

General comments

Q18. As in the review is mainly based in vitro studies, authors could add a paragraphe to explain the differences and application of melanoma cell lines mentioned in the review to help readers to better understand the in vitro reports. 

Q19. Authors  must indicate in the results or disuccion section if the anti-cancer effects observed for extracts or pure natural products are selective or not against non-cancerous cell lines. 

Q20. For most promising natural products, authors can provide their chemical structures. 

Comments on the Quality of English Language

Some phrases can be improved. 

Round 2

Reviewer 1 Report

Comments and Suggestions for Authors

I recommand it for publication. 

Author Response

Thank you for your feedback. We appreciate the thorough evaluation and have carefully addressed the concerns raised by Reviewer 2.

Q15: We revised the discussion section to better highlight the differences in activities (apoptosis, anti-metastasis, antioxidant, and anti-angiogenesis) and emphasized the potential of natural products as adjuvants in melanoma treatment.

Q16: We clarified the available in vivo studies regarding acetoside and melanoma, ensuring that our review accurately reflects the current state of research.

Q18: A paragraph explaining the differences and applications of melanoma cell lines was added to aid in the interpretation of in vitro results.

Q20: We have incorporated Figure 5, which presents the chemical structures of key natural compounds (Curcumin, Quercetin, Acteoside, Cynaropicrin, and Hinokitiol), as requested. Additionally, we revised the discussion section to explicitly refer to this figure.

We believe these modifications fully address the concerns raised. The revised manuscript is attached, with all changes highlighted in yellow for clarity. Please review the attached file, and let us know if any additional modifications are required.
